# A Review of Electrospun Nanofiber Interleaves for Interlaminar Toughening of Composite Laminates

**DOI:** 10.3390/polym15061380

**Published:** 2023-03-10

**Authors:** Biltu Mahato, Stepan V. Lomov, Aleksei Shiverskii, Mohammad Owais, Sergey G. Abaimov

**Affiliations:** Center for Petroleum Science and Engineering, Skolkovo Institute of Science and Technology, Bolshoy Boulevard 30, bld. 1, Moscow 121205, Russia

**Keywords:** fracture toughness, electrospun veil/interleave, delamination, toughening mechanism, cohesive zone modeling

## Abstract

Recently, polymeric nanofiber veils have gained lot of interest for various industrial and research applications. Embedding polymeric veils has proven to be one of the most effective ways to prevent delamination caused by the poor out-of-plane properties of composite laminates. The polymeric veils are introduced between plies of a composite laminate, and their targeted effects on delamination initiation and propagation have been widely studied. This paper presents an overview of the application of nanofiber polymeric veils as toughening interleaves in fiber-reinforced composite laminates. It presents a systematic comparative analysis and summary of attainable fracture toughness improvements based on electrospun veil materials. Both Mode I and Mode II tests are covered. Various popular veil materials and their modifications are considered. The toughening mechanisms introduced by polymeric veils are identified, listed, and analyzed. The numerical modeling of failure in Mode I and Mode II delamination is also discussed. This analytical review can be used as guidance for veil material selection, for estimation of the achievable toughening effect, for understanding the toughening mechanism introduced by veils, and for the numerical modeling of delamination.

## 1. Introduction

Fiber-reinforced composite materials have excellent mechanical properties, corrosion resistance, and creep resistance compared with traditional materials [1,2,3], and consequently, are widely accepted and used for various structural applications in the aircraft, automobile, energy, ship, civil, sports, and offshore industries, to name a few. Such high-performance structural composite laminates are commonly produced either using autoclave technology or using liquid composite molding, based on preforms as layups of 2D plies with fibrous reinforcement. These laminates have high in-plane mechanical properties that are determined by the fibers, but suffer from low out-of-plane properties because interlaminar fracture toughness (FT) is only provided, besides the matrix, by partial fibrous involvement in the form of the fiber bridging effect. This makes a composite laminate highly susceptible to failure under through-the-thickness loads and out-of-plane low-velocity impacts. Events such as matrix and fiber cracking, as well as delaminations, are observed during such failure. The low interlaminar FT of a composite laminate remains one of the limiting factors during its service life. Poor interlaminar strength and interlaminar FT are thus major limitations of fiber-reinforced composite laminates.

The concentration of high interlaminar shear and transverse stress near the edges, possible pre-cracks or manufacturing defects, points of laminate curvature, and drilled holes are some of the possible causes of delamination initiation. In these cases, failure occurs via both modes (Mode I and Mode II) of failure, resulting in various in-service problems. Various methods are used to improve the interlaminar properties of structural composites: matrix toughening [4,5], 3D reinforcement (3D weaving [6], stitching [6], Z-pinning [7]), nano-stitching [8], and fiber hybridization [9]. These methods improve the FT but come at the expense of either escalated complexity, increased cost/weight, or the loss of in-plane properties. For example, the involvement of 3D reinforcement, in the form of 3D weaving, stitching, or Z-pinning, solves the problem but cannot be implemented in structures with high load-carrying performance because of the fiber crimp and, thereby, loss in the targeted stiffness/mass ratio. In contrast to the 3D reinforcement method, nano-stitching improves FT and does not degrade the in-plane properties, but it is not a scalable method. Therefore, industry applications require measures to improve interlaminar FT in laminates (as layups of 2D plies), which would overcome the aforementioned drawbacks.

Recently, the use of polymeric non-woven nanofiber in the form of a thin mat has been a popular approach to toughening composite laminates. In this method, the thin mat is introduced as an additional layer between the laminae of a composite laminate. The thin mat is commonly known as a veil/interleave, and the method of introducing the veil/interleave is known as interleaving. The fiber diameter in the fibrous veil/interleave ranges from tens of nanometers to a few micrometers. The fine diameter and evenly distributed fibers of the veil ensure low areal density and low thickness [10]. Hence, the impact of introducing a veil/interleave on the laminate thickness and mass is negligible. The overall fiber volume fraction in the composite laminate is not affected much, guaranteeing the lowest level of compromise on the in-plane mechanical properties of the laminate. Similarly, the veil/interleaves are highly porous, and thus, do not disrupt the resin flow during impregnation or curing [11]. The effectiveness of veils/interleaves in toughening composite laminates has been demonstrated without doubt [12,13,14]. Melt blowing, solution blowing, electrospinning, etc., are some methods of producing such non-woven polymeric veils.

Four parameters were used to quantify and characterize the effectiveness of toughening, measured in terms of FT for the initiation and propagation of Mode I and Mode II delamination. These parameters are extensively reported for various fiber, matrix, and interleave systems of laminates. The toughening effect depends on the veil material, thickness, fiber diameter, areal density, form (melted vs. non-melted, solid vs. hollow), reinforcing fibers, and compatibility amongst the veil, fiber, and matrix. Most of the literature on this topic (see the publication statistics below) has been partially summarized in several reviews [15,16,17], with the most recent published in 2017.

This analytical review focuses on the toughness efficiency of the veils/interleaves produced via electrospinning. It offers a systematic comparative analysis of data from the literature up to 2022 from the viewpoint of the targeted increase of interlaminar FT using interleaving polymeric veils. Both modes of delamination failure observed in the baseline and interleaved composite laminates are covered for various veil materials. The toughening mechanisms added through the introduction of veils are listed, analyzed, and discussed. The numerical modeling of failure by both modes of delamination is also discussed. The present analytical review can be used as guidance for veil material selection, for estimation of the achievable toughening effect, for understanding the toughening mechanisms introduced by veils, and for the numerical modeling of delamination.

Section 2 presents the methodology followed for data gathering and analysis. It also shows the statistics and trends of publications on the topic of the review. Section 3 describes the electrospinning process used for the production of veils and briefly explains its history. Section 4 presents a comparative analysis of the toughening effects achieved by interleaving electrospun veils manufactured from different polymers. Section 5 analyzes the toughening mechanism noted for electrospun veil-interleaved laminates. Section 6 outlines a statistical analysis of the collected data on the basis of mode ratio and areal density. Section 7 introduces a general methodology for developing a numerical model of the toughening effect of electrospun veils using cohesive zone modeling. Finally, conclusions are drawn in Section 8.

## 2. Methodology

The factual data on toughening were structured based on the veil material (polyethylene terephthalate (PET), polyphenylene sulfide (PPS), polyamide (PA), polyacrylonitrile (PAN), and polycaprolactone (PCL)), including their hybrids and modifications, such as metal-, CNT-, or graphene-modified veils. The structured data contained information on reinforcement, the matrix and veil, the production methods, the FT of the baseline and interleaved laminates for Mode I and Mode II, improvements in the FT of interleaved laminates compared to the baseline laminate, and identification of their application area and toughening mechanics. The collected factual data were used for analysis and can be obtained from http://bit.ly/3TEDvzn (accessed on 7 March 2023). The analysis was carried out using MS Excel^TM^ 2010 and MATLAB^TM^ R2021.

Publications derived from keywords related to the topic, such as “veil OR electrosp*” (herein referred to as VE), “VE” AND “composite OR laminate” (herein referred to as CL), and “VE” AND “CL” AND “toughness”, were searched in the Web of Science database (core collection, all years up to September 2021). This revealed the ongoing dynamics in the field’s development. The search was further narrowed using the filter “fiber OR fibre”. The results are shown in Figure 1; note the logarithmic scale on the vertical axis in Figure 1a.

“VE” as a keyword for the search provided extensive results. However, the narrowing to fibrous materials helped to focus the search from 117,862 to 15,621 documents, including articles, conference papers, reviews, book chapters, and editorial materials. Similarly, there were 5272 and 240 documents on “VE AND CL” and “VE AND CL AND toughness”, respectively, when narrowed using the filter “fiber OR fibre”, as presented in Figure 1a. The data show escalating interest in the proposed topic, which grew fast in the last decade, with the number of publications on “VE AND CL AND toughness” increasing at a rate of 10x. This increase reflects general growth in publication activity. Therefore, the numbers were compared with the total number of publications on the toughness of fiber-reinforced composites (“Composite AND Toughness” refined using “fiber OR fibre”) to calculate the percentage contribution of VE as a toughening technology. The total contribution increased from below 1% in 2001 to ~6% in 2022, as presented in Figure 1b. A linear fit to the data shows a steadily increasing trend of publications in the field. So, the topic of the present research is in the growth phase of the technology evolution curve.

**Figure 1 polymers-15-01380-f001:**
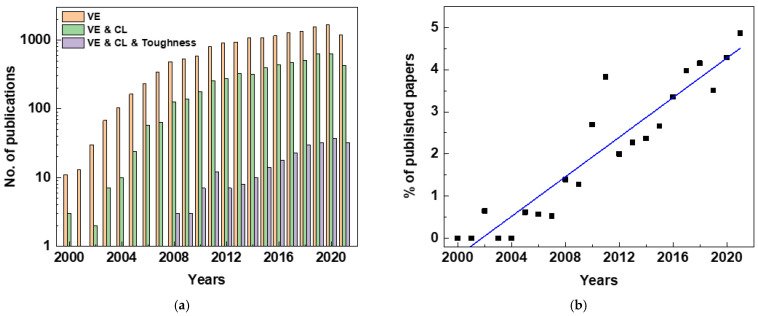
Statistics of publications on electrospun toughening veil interleaves: (**a**) various keywords (log scale on vertical axis); (**b**) growing trend of using VE for toughening composite laminate: articles on use of VE as a proportion of all papers on toughening FRP composite laminates.

## 3. Production of Veil

Electrospinning is a technology that is more than a century-old [18]. However, applications for laminate interleaving appeared much later [17], and the US patent, generally accepted as the pioneering invention, was awarded in 1999 to Y.A. Dzenis and D.H. Reneker [19]. The patent focused on interleaving an “electrospun sheet” as a veil between two plies of a composite laminate to improve its mechanical performance.

Electrospinning is a flexible, simple, and cost-effective technology that is used to produce extremely fine fibers for a wide range of materials, with diameters ranging from tens of nanometers to a few micrometers [18]. It is a top-down technique for manufacturing, [20] where the millimeter-sized polymer pellets are dissolved in an organic solvent, and then, electrospun. Electrospun nanofibers are long, continuous, easily aligned, and inexpensive. These nanofibers have unique properties, such as high surface area-to-volume ratios, high aspect ratios (length/diameter), and high mechanical properties (stiffness and strength) because of the high molecular orientation along the fiber axis. Electrospinning can easily be scaled for mass production in industrial applications [18,20,21].

Figure 2a shows a schematic representation of electrospinning in the manufacture of an electrospun veil. The major parts of the setup include a syringe with a nozzle at its tip, a conducting collector plate, and a high-voltage source that connects the collector and the nozzle. The syringe with the nozzle tip holds the polymer solution. When high voltage is applied, the polymeric solution in the syringe is pulled out of the syringe. Liquid droplets are formed at the tip of the nozzle, which are further converted into a jet of polymeric liquid, finally being collected on the conducting collector plate; this results in the formation of a continuous polymeric fiber. After collecting layers of this continuous fiber, one over the other, it forms a non-woven, porous nano-fiber veil (Figure 2b). The diameter of the fiber depends on the applied voltage and polymeric solution. Similarly, the veil’s areal density and thickness depend on the duration of manufacturing. Once the desired thickness is achieved, the electrospun veil can be separated from the collector and transferred onto a substrate.

The electrospun veils are placed between two plies of fibrous reinforcement at desired locations in the laminate layup. Then, the fiber-reinforced polymer (FRP) laminate is manufactured as per the standard manufacturing procedure. Manufacturing methods include vacuum infusion, compression molding or press-clave, autoclave, and hand wet-layup followed by vacuum bagging, to name a few. The veil thickness under the processing pressure is sufficiently small. Hence, the laminate’s fiber volume fraction is not affected much, and the laminate’s in-plane mechanical properties are preserved [10].

Depending on the test procedure, the samples used to measure toughness are manufactured with the interleaves placed either at the laminate mid-plane or at every ply/ply interface. In particular, to measure the Mode I initiation and propagation energy, a DCB test is conducted as per the ASTM D5528 standard [22]. Similarly, the ENF test measures the Mode II initiation and propagation FT per the ASTM D7905 standard [23].

## 4. Comparative Analysis of the Toughening Effect

The FT values reported in the literature were analyzed based on the veil materials and their modifications, and a comparative analysis of attainable FT with the veil was conducted. The results are discussed below.

### 4.1. Polyethylene Terephthalate

Tzetzis et al. [24,25], Kuwata et al. [26,27], Quan et al. [14,28,29], Fitzmaurice et al. [30], and Del Saz-Orozco et al. [31] studied the effect of interleaving a PET veil in Mode I and Mode II on the toughening of glass fiber and carbon fiber laminate composites. A general trend of improvement in all the FT parameters was observed, with an exception noted in Mode I properties by Del Saz-Orozco et al. [31].

#### 4.1.1. Neat PET Veil

Tzetzis et al. [24,25] explored the possibility of using a veil as an interleave for repair purposes (to attach patches) in GFRP laminates [24,25]. Significant improvements in G1i and G1p were reported with increasing veil areal density, reaching up to 740% and 770%, respectively, at a veil areal density of around 25–40 g/m^2^. However, it should be noted that these significant improvements were relative to the values reported for the “as-received” in-service GFRP surfaces. When these “as received” in-service laminates were treated using hand abrasion and grit blasting, the toughness increased significantly, even in the absence of interleaves. The improvement due to veil interleaves can be re-calculated relative to “treated” surfaces, after which it becomes only moderate of 16–49%.

In contrast, Kuwata et al. [26,27] explored the effect of a PET veil as an interleave in a newly manufactured CFRP laminate. The study investigated the veil’s effect on various epoxy matrices (in civil and aerospace applications) and CF reinforcement architectures (UD, satin weave, and plain weave). For the UD-reinforced composite, a stable but moderate increase of 15–56% was observed, which occurred almost independently of resin type. The low value of this increment is probably explained by the interleave replacing the intrinsic UD fiber’s bridging effect with the tough thermoplastic bridging of nanofibers, thus replacing the strong UD plie bridging. For woven composites, the effect of PET interleaves was stronger, as expected, due to the weak fiber bridging of the original reinforcement. For the satin weave composites, the PET veils had a moderately positive effect (up to 94% for epoxy; up to 33% for vinyl ester) on Mode I toughness and a very pronounced effect on Mode II toughness, increasing the values to 3100–3630 J/m^2^ for initiation and 3750–4760 J/m^2^ for propagation, with the effect being stronger for epoxy comparing to vinyl ester. For the plain weave composites, the veils’ introduction generated a significant positive effect only for epoxy resin (Mode I: up to 175%, Mode II: up to 88%), while for vinyl ester, the effect, although always positive, was less significant. A similar effect of moderate improvement in the UD CF reinforcement but stronger improvement in the weave CF reinforcement (5H and NCF weave) was also noted by Quan et al. [14,29]. The values of FT noted by Kuwata et al. [26,27] and Quan et al. [14,29] are in a comparable range despite slight differences in their weave architectures (satin and plain weaves vs. 5H and NCF weaves).

Similarly, Fitzmaurice et al. [30] studied the effects of different numbers of veil layers in a laminate (one vs. two veil layers between each pair of plies) in weave GF reinforcement, and noted a crack deviation from the veil region to the fiber–matrix interface region. Similar crack deviation was also noted by Del Saz-Orozco et al. [31] in UD GF reinforcement. The phenomenon of crack deviation was attributed to a strong interaction between the PET veil and the matrix. In spite of crack deviation, Fitzmaurice et al. [30] noted moderate improvement in FT, but Del Saz-Orozco et al. [31] noted a decrease in FT because of the difference in the GF reinforcement architectures (weave vs. UD).

#### 4.1.2. PET-CF Hybrid Veil

Tzetzis et al. [25] repaired an in-service FRP structure by grit blasting the surface before the CF or PET-CF hybrid veil was inserted as an interleave. It was inserted between the repaired structure and the patch, followed by infusion. Grit blasting of the in-service FRP structure induced many new sites for stronger bonding between the structure and the patch material. The implemented veil did not impede the repair and enhanced the fiber bridging effects. Great improvements (up to 600% for initiation and 1100% for propagation) were reported for Mode I toughness, but these were mainly attributed to grit blasting.

The use of CF or PET-CF hybrid veils in newly manufactured laminates had a mixed effect [26,27]. For satin weave composites, the detrimental effect of CF implementation was masked by the improving toughness induced by PET. Still, the tendency was clear: the higher the share of CF implemented, the poorer the results were. A pure CF interleave led to a decrease in all values of FT. For the plain weave laminate, the tendency was not pronounced but still present. CF interleaves improved Mode II toughness for the UD laminates, but their use cannot be recommended because of the degradation of the Mode I toughness properties.

#### 4.1.3. Nano-Modified PET Veil

PET veils have demonstrated their toughening capabilities; however, introducing a PET veil decreases a laminate’s electric conductivity [28]. To address this problem, two different concentrations of MWCNTs were airbrushed on a PET veil surface. It was found that airbrushing as little as 0.4 g/m^2^ of MWCNTs on the veil improved the laminate’s FT (up to 65% for Mode I and 100% for Mode II) and overall electrical conductivity (up to 65%), which is a significant improvement in FT and electrical conductivity compared to the baseline laminate properties, whereas the neat PET veil improved FT only. Airbrushing a higher concentration of MWCNTs increased the electrical conductivity further but decreases the FT.

In summary, PET veils have a positive impact on the toughening mechanism in both modes for both epoxy and vinyl ester resins, for both UD and weave CF reinforcement, and for weave GF reinforcement architectures. The effects with epoxy are typically much more pronounced than with vinyl ester. PET-CF hybrid veils show mixed results. Airbrushing a lower concentration of nano-reinforcement on the veil surface is enough to improve the FT and electrical conductivity. Based on these results, a PET veil and its nano-modified counterpart are recommended for structural delamination control applications.

### 4.2. Polyphenylene Sulfide

Quan et al. [14,29,32], Ramirez et al. [33], and Ramji et al. [34] reported the application of a PPS veil to improve the fracture performance of a CFRP composite. No studies have been reported for GFRP. Quan et al. studied the effect of an interleaving neat PPS veil [14,29] and a nano-modified PPS veil [32] on FT. The neat PPS veil was interleaved for different CF reinforcement architectures (UD vs. weave), whereas the nano-modified PPS veil was interleaved for UD CF only. Ramirez et al. [33] explored the impact of the manufacturing directionality (compared to reinforcing UD CF) of the veil on FT. Ramji et al. [34] focused on the combined effect of interleaving the PPS veil and the interfacial orientation of CF on delamination migration and FT.

#### 4.2.1. Neat PPS Veil

Interleaving the neat PPS veil in the CF laminate showed a clear trend of high FT for weave CF reinforcement in comparison to UD CF reinforcement [14,29]. A similar trend was also observed for the neat PET veil. This trend was observed to be moderate for Mode I (up to a ~75% increase) and profound for Mode II loading, increasing the value of FT to 2200–2600 J/m^2^ for initiation and 3000–3200 J/m^2^ for propagation.

Ramirez et al. [33] manufactured CF laminates with PPS veils oriented in the manufacturing direction (MD) and in the cross-direction (CD), perpendicular to the MD. A significant change (up to ~2x) in Mode I FT was noted for a thick veil of 38 g/m^2^ irrespective of veil orientation. It is necessary to note that the mechanical anisotropy of the PPS veil showed no impact on the achievable FT because a PPS veil is a non-woven structure with randomly oriented fibers.

Ramji et al. [34] observed delamination migration when an element of 90° or 45° plies was present in the midplane. Delamination migration added additional crack propagation sites, leading to high FT. It can be noted that the highest FT value was reported for 90°/90° compared to all other interfacial orientation combinations of 0°, 45°, and 90°.

#### 4.2.2. Nano-Modified PPS Veil

Higher improvement in the FT was noted when the neat PPS veil was modified through doping with a lower concentration (up to 0.6 g/m^2^) of MWCNTs in both Mode I and Mode II. However, when MWCNT concentration was increased to 1.45 g/m^2^, the Mode I FT decreased, becoming worse than for baseline laminates. The Mode II FT also decreased only moderately, but was still better than the baseline. Irrespective of concentration, doping GNP on the neat PPS veil showed the detrimental effects on Mode I FT. A lower concentration of GNP showed good improvement in Mode II FT, which decreased slightly at a higher concentration.

Such behaviors of MWCNT and GNP can be attributed to their 1D and 2D shapes, respectively. The addition of a small wt.% of MWCNTs introduced additional interactions, such as MWCNT pull-out and breakage between the PPS nanofibers and the epoxy matrix. This resulted in improved PPS fiber/epoxy adhesion, and subsequently, led to additional PPS fiber breakage and an improved nanofiber bridging mechanism as a toughening mechanism during the fracture process. For these reasons, the fracture energy was further increased by doping a small amount of MWCNTs on the PPS veils. However, at a high wt.% of MWCNTs, the PPS nanofiber/epoxy adhesion increased to a sufficient level to prevent the PPS nanofiber pull-out and fiber bridging (the primary toughening mechanism). This caused a considerable drop in fracture energy. In contrast, the 2D-structured GNPs significantly agglomerated and attached to the PPS nanofibers. This resulted in a decline in PPS nanofiber/epoxy adhesion [32].

To sum up, interleaving a neat veil improves laminate fracture performance irrespective of the fiber architecture; however, higher improvement was noted in the weave CF compared to the UD CF. The airbrushing of a lower concentration of MWCNTs on the veil surface improves the fracture performance, as well as the electrical conductivity, of a laminate. Airbrushing GNPs increases electrical conductivity but decreases the FT. It is noteworthy that a higher concentration of MWCNTs and all concentrations of GNPs also cause drastic drop in FT. Based on these results, neat and low-concentration MWCNT doping of PPS interleaves is recommended for delamination control in structural applications. Doping nano-reinforcement using materials such as MWCNTs and GNPs improves electrical conductivity; hence, it can be recommended for lightning strike protection, electro-magnetic shielding, and damage detection applications.

### 4.3. Polyamide

Beckermann et al. [11], Meireman et al. [12], Garcia-Rodriguez et al. [13], Quan et al. [14], Kuwata et al. [26,27], Del Saz-Orozco et al. [31], Saghafi et al. [35,36], Nash et al. [37,38], Guo et al. [39], Chen et al. [40], Ognibene et al. [41], Pozegic et al. [42], Beylergil et al. [43,44,45], Alessi et al. [46], Barjasteh et al. [47], Monteserin et al. [48,49], Daelemans et al. [50,51,52], De Schoenmaker et al. [53], O’Donovan et al. [54], and Hamer et al. [55] reported that PA veils as interleaves for FRP composite laminates are applicable in either carbon or glass fiber reinforcement. PAs of different classes were used in these studies, such as PA 6 (sometimes, the brand name Nylon is used), PA 11, PA 12, PA 66, and PA 69. PA is the most popular material for veils amongst the presented materials, as demonstrated by the higher number of studies on PA veils. It has been extensively considered as an interleave for composites manufactured via not only VI and VARTM, but also compression molding and autoclave manufacturing methods.

#### 4.3.1. Neat PA Veil

A general trend of improvement in the overall fracture performance of an FRP laminate has been observed for both CF and GF, with the exception of decreases in Mode I FT [23,24,34,39,43,47,50]. These exceptions are caused by various factors, such as compatibility [26,27,37], mesoscale inhomogeneity due to either a thick veil [53] or a thick veil fiber [46], and a weak interface [42]. PA veils in CF reinforcement have been identified as more compatible with vinyl ester-based matrixes than epoxy-based matrixes [26,27]. Amongst the epoxies, the PA veil is more compatible with BZ9120 epoxy than BZ9130 epoxy reinforced with CF [37].

Quan et al. [14], Kuwata et al. [26,27], and Daelemans et al. [47] observed high FT for weave CF reinforcement and low FT for UD reinforcement interleaved with a PA veil. Similar trends are also noted in PET- and PPS-interleaved veils. Woven reinforcement showed promising improvements due to the fabric architecture. This resulted in plastically failed PA nanofibers zones, indicating good load transfer to the nanofibers. Comparatively, in UD reinforcement, the nanofibers blocked the formation of the carbon fiber bridging zone and delamination propagation between the nanofibers. This resulted in relatively low improvements in FT.

#### 4.3.2. Metal-Modified PA Veil

PA veils are modified with different kinds of metal-based chemicals for various purposes, which include altering the stiffness and hardness of nanofibers [11], adding antibacterial effects [48], and developing electrically conductive laminates [39,56]. Modification techniques include precipitating AgNO_3_ throughout the nanofiber veil [11], dispersing TiO_2_ nanoparticles on the nanofiber veil [48], painting AgNW solution on veil surface [39], and coating Ag (silver-based salt) via electroless plating of the veil [56].

Interleaving a modified PA veil with a AgNO_3_ coating [11], AgNW painting [39], a pure Ag paste coating [56], and TiO_2_ nanoparticle dispersion [48] demonstrated some improvement in fracture performance. These studies show that improvement in the modified conditions was comparatively lower than for the neat PA-interleaved laminate; however, it was still higher than for the baseline laminate. Such slight deterioration in FT improvement for the modified veil was caused by a weak interface formed by the metal surface. However, it should be noted that AgNWs have the strongest interface amongst them.

An Ag-modified veil adds the multifunctionality of improved conductivity to a laminate. For instance, AgNW-painted veils improve laminate conductivity by up to 100× in-plane and by 10× in the thickness direction [39]. Similarly, Ag-plated veils improves conductivity by 1500× in-plane and by 25× in the thickness direction [56]. TiO_2_ adds antibacterial functionality, making the laminate suitable for marine applications [48]. There is strong, growing interest in the development of multifunctional interleaves. Metal-modified veils can be used as multifunctional interleaves, but slight deterioration in FT must be considered. The multifunctionality of upgraded conductivity and improved antibacterial properties can be successfully achieved; however, altering the stiffness and hardness of a PA nanofiber with a AgNO_3_ coating has not been achieved [11].

To sum up, the interleaving of a PA veil always increases Mode II FT, irrespective of the reinforcement type or the binding matrix. However, for Mode I FT, improved fracture performance depends on various factors, such as the compatibility of the PA veil with the binding matrix, crack path, reinforcement type, and areal density, or the thickness of the veil. Based on these results, the PA veil is recommended for delamination control in FRP laminates for structural applications. Similarly, metal-modified PA veils are also recommended for the multifunctionality of laminates.

### 4.4. Polyacrylonitrile

VanderVennet et al. [57], Zhang et al. [58], Chiu et al. [59], Razavi et al. [60], Molnar et al. [61], and Eskizeybek et al. [62] studied a PAN interleave’s effect on CFRP laminates and its effect on FT. Zhang et al. compared a PAN veil to other veils, whereas others have studied the PAN veil and its nano modifications.

#### 4.4.1. Neat PAN Veil

The neat PAN-interleaved laminates showed a significant depreciating effect of up to −70% in G1i compared to the baseline laminate [57,58,59,62]. They were outperformed by veils of other material, including doped PAN [62]. This reduction in toughness was mainly attributed to problematic impregnation in the presence of a dense veil. A minor positive effect in G1i, with an increase of around 18%, was observed only in two cases: for VARTM [60] with very low veil areal density of 1 g/m^2^, and the autoclaving [57] of prepregs, where, for both cases, impregnation does not present a problem. The effect of neat PAN interleaves on G1p was explored, and a mixed result of 15% deprecation [58] and 22% improvement [62] was found.

#### 4.4.2. Nano-Modified PAN Veil

Some examples of nano-modifications include nanoparticles of Al_2_O_3_ [60] and CNT [61,62], mixed in PAN’s electrospinning solution, i.e., incorporated into the veil fiber. Compared to the baseline laminate, nano-modified PAN veils showed better overall performance depending on the wt.% of the nano-reinforcement. Al_2_O_3_ nanoparticles improved FT by up to 47%, and CNT improved FT by 6–27% in G1i and 45–77% in G1p [62]. Similarly, introducing CNT to an electrospun PAN fiber also improved the electrical conductivity (up to 50%) and thermal conductivity (~3x) of the laminate [61].

To summarize, neat PAN interleaves generally show minor improvement but can be detrimental if impregnated with viscous resin. However, doping nano-reinforcements, such as Al_2_O_3_ or CNT, demonstrates significant improvement, as well as providing multifunctionality. This merit can probably be attributed to the nano-reinforcement delivered to the ply/ply interface via veil placement rather than via PAN’s direct involvement.

### 4.5. Polycaprolactone

Beckermann et al. [11], Saghafi et al. [32], Cohades et al. [63], and Heijden et al. [64] reported the effect of PCL veils on G/CFRP laminates. A similar trend of higher FT for woven reinforcement has also been noted for PCL interleaves.

Beckermann et al. studied several veil materials and found a correlation between Mode I FT and the ultimate elongation of the bulk polymer used to make the veil. PES, PAI, PA66, and PVB followed this trend, but PCL did not, despite it having the highest ultimate elongation (679–948%). Upon further investigation, it was found that the PCL veils melted during oven curing of the laminate due to their low melting temperature. Thus, the primary toughening mechanism (plastic deformation-identified) was replaced by a phase-separated microstructure, leading to a slight improvement in toughness (up to 14%). Similarly, a phase-separated microstructure where epoxy particles are surrounded by a PCL matrix was identified as a toughening mechanism by Saghafi et al. and Cohades et al. Similarly, Beckermann et al. also noted a correlation between the Mode II FT and the tensile strength of the bulk polymer used to manufacture a veil. This correlation was followed by all the veils investigated. The tensile strength of PCL was low compared to that of PA66 (9 MPa vs. 85 MPa); hence, a small improvement of 7% in Mode II toughness was noted in PCL veil-interleaved laminate (compared to 29–69% for PA66 veil). Similarly, a 24% improvement in PCL (vs. 68% in PA66) was also noted in [35]. It is noteworthy that the authors did not describe the effect of PCL melting in Mode II toughness.

Such atypical behavior of the PCL veil was further investigated, and the authors obtained up to 94% improvement in Mode I fracture toughening [64]. It was noted that a room-temperature pre-cure (before oven curing at 80 °C) of laminate eliminates such atypical behavior and acts as a crucial step when the PCL veil is interleaved. The impact of such a step has not been explored in Mode II experiments. It is worth mentioning that Heijden et al. used single- and double-layered PCL as interleaves, increasing the net areal density of the veil.

Cohades et al. assessed the possibility of using a PCL veil for toughening and healing cracks. Thermal treatment at 150 °C was applied for 30 min to the cracked specimens to assess PCL’s capability to bleed, flow, and bridge the cracked faces, thus healing the cracked laminate. However, such healing was not observed because of the high viscosity of the high-molecular-weight PCL, which was used in this study, with a required healing time of around 100 h for the fine nanostructure of the pores (compared to the 30 min applied in tests). The authors concluded that self-healing properties could be achieved either by increasing the diameters of the nanofibers, which may compromise the FT, or by decreasing the viscosity with the application of the low-molecular-weight PCL. However, in this case, the electrospinning becomes unstable due to low molecular entanglement.

These analyses show that PCL interleaves provide only mild improvements compared to other polymers, so they are not recommended for structural applications. Although a room-temperature pre-cure removes atypical PCL behavior, such effects are not studied to Mode II.

To summarize the comparative analysis, it is clear that the introduction of a polymeric nanofiber veil improves the overall FT, with some exceptions. A comparison of attainable FT upon interleaving these veils is plotted in Figure 3, which shows cloud point plots of four FT parameters, plotted as the Mode I vs. Mode II initiation FTs in Figure 3a, and the Mode I vs. Mode II propagation FTs in Figure 3b. Mode lines are also added, which is discussed separately in Section 6. This plot can be used as a reference plot for material selection as per the FT requirement in the design of composite structures in engineering applications. For instance, for composite design anticipating a higher G1i and a moderate G2i, a PA veil is recommended. Similarly, for a higher G2i and a moderate G1i, a PET veil is recommended.

## 5. Toughening Mechanism

Cracks propagate in FRP laminates through the epoxy-rich areas, such as the interlaminar region, resin pockets, etc. The brittle nature of the epoxy aids in such crack propagation. However, interleaving the laminates with veils results in non-linear fracture patterns. This change is brought about by several toughening mechanisms added by the veil to the resulting laminates. Nanofiber bridging or crossings, nanofiber pull-out and debonding, nanofiber plastic deformation and breakage, crack pinning, crack deflection, strong adhesion bonds between the resin and veil due to compatibility or melting of the veil, and fusion-bonded dots are the toughening mechanisms that have been identified. (Fusion-bonded dots are semi-spherical dots formed by melting the veil at regular locations within the veil, which reduces crack propagation and improves the FT.) Some of these toughening mechanisms are shown in Figure 4. Doping veils with nano-reinforcements complicates the aforementioned behavior, mostly by changing the veil/resin adhesion, delivering nano-reinforcement to the fracture zone, and modifying the functional properties of the laminate, such as conductivity.

The nanofiber bridging or crossing formed by the nanofibers that compose the veil has been identified as the most common toughening mechanism. The nanofibers bridge the two laminae on the opposite sides of the veil, preventing crack initiation and delaying propagation. Upon further loading, the nanofiber is pulled out of the epoxy, inducing debonding. This is followed by nanofiber plastic deformation and nanofiber breakage. These toughening mechanisms consume much energy, resulting in an increase in the overall FT of the interleaved laminates. The effectiveness of the nanofiber bridging toughening mechanism depends on proper load transfer to the nanofibers. Crack propagation under Mode II loading results in much higher improvements than Mode I loading due to the alignment of the loading with the nanofiber direction in the veil plane. In Mode I crack propagation, the loading of the nanofibers is less optimal and is shown to be dependent on both the primary reinforcement fabric architecture and the presence of a reinforcement fiber bridging zone.

**Figure 4 polymers-15-01380-f004:**
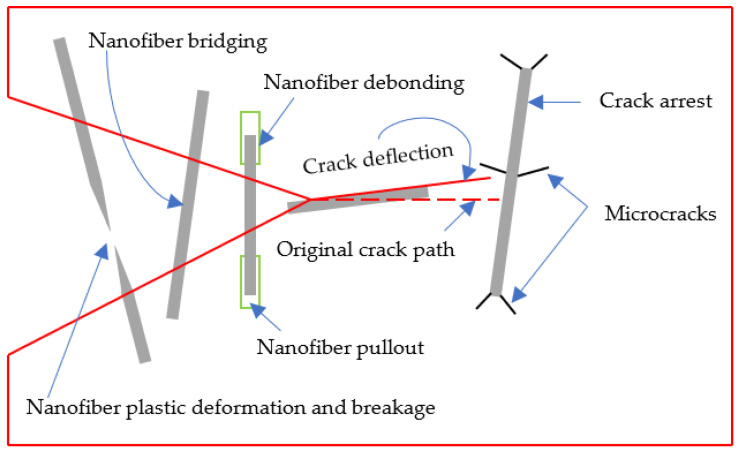
Schematic diagram of the toughening mechanism identified in interleaved laminates.

Crack pinning and crack deflection are also observed as toughening mechanisms that consume energy to improve toughness. Crack deflection and delamination migration occur when a strong adhesion bond is formed between the lamina and the veil due to material compatibility or ply orientation. Material compatibility can be observed in its neat form or through phase changes noted during curing for meltable veils.

## 6. Statistical Analysis

### 6.1. Mode Ratio

The mode ratio $\left(m=\frac{G2}{G1}\right)$ inherits the nature of the resin used in the composite. The introduction of an electrospun veil changes the overall behavior of the resin in the interlaminar region. For a baseline laminate, a low mode ratio close to unity is expected for ductile resins. In contrast, a much higher ratio is observed for brittle resins. The change in the ratio due to interleaving depends on the changes in the individual properties, which further depends on the type of veil, its areal density, and its production method, as noted in Section 3.

Figure 4 shows the initiation and propagation FT of the baseline and interleaved composite laminates. The scatter of points for baseline vs. interleaved laminates indicates a change in the nature of the resin at the interlaminar region due to the addition of the electrospun veil in the interlaminar region of the laminate. The FT of baseline laminates is concentrated near the bottom left corner of the plot, in contrast to the FT of interleaved laminates, which is distributed throughout. The highest Mode I initiation and propagation FTs are noted for laminate interleaved with the PA veil, whereas the highest Mode II initiation and propagation FTs are noted for laminate interleaved with the PPS-and-PET-CF hybrid veil, respectively.

Similarly, adding mode ratio (m = 1 … 32) lines shows some other characteristic behaviors. For instance, the initiation FT of baseline laminates lies between mode ratios of 1 to 16, and most of the values are concentrated around a mode ratio of 4, whereas the propagation FT of baseline laminates lies between mode ratios of 1 to 8, and most of the values are concentrated between mode ratios of 2 and 4. Similarly, in the case of interleaved laminates, the mode ratio reaches up to more than 32 for the initiation FT and more than 16 for the propagation FT.

### 6.2. Areal Density

Further, an areal density of zero indicates the baseline laminates, and a non-zero areal density indicates that the laminate is interleaved with a veil. The minimal, mean, and maximal areal density values are 0, 10, and 40 g/m^2^, respectively. The veil’s areal density and FT, as presented in Table 1, show a positive correlation between the four FT parameters and areal density. The mode II FT is more strongly correlated with the areal density (*r* > 0.5) than the Mode I FT (*r* < 0.5). Figure 5 plots a cloud of points of the Mode I and Mode II initiation and propagation FTs for UD CF laminates when interleaved with an electrospun veil produced with different materials and areal densities. The denser cloud of points in Figure 5a,c shows that extensive research has been carried to study the interleaving effect on initiation energy. However, only limited research has focused on measuring propagation FT (applicable to both modes). The maximal attainable value of G1i is ~1600 J/m^2^, of G1p is ~1600 J/m^2^, of G2i is ~7000 J/m^2^, and of G2p is ~6000 J/m^2^.

PA has been identified as the most common choice of polymeric material for interleaving based on the reported number of data points, as shown in Figure 5. Various modifications of these materials are also used. These modifications include coating/doping with nano-reinforcement, such as CNT, GNP, metal modification/plating, or mixing one polymeric material or CF with another. Such modifications of the veil add a new feature, making the veil multifunctional. A broad range of areal densities has been investigated for PA interleaves, showing a stable interleaving effect on improving FT. However, such a wide range of studies is missing for other veil materials.

## 7. Numerical Modeling of Delamination

Numerical modeling is typically carried out using a 2D strain finite element model to simulate the DCB and ENF beam while cohesive elements are used to model the delamination behavior. “Traction” and “Separation” at the interface with the possibility of delamination by crack propagation exist, and they govern cohesive element modeling. The cohesive elements initially behave linearly until a threshold level, usually known as the damage initiation point, is reached. The stiffness and strength decrease progressively upon reaching the initiation point, which is known as damage evolution or softening. This continues until the surfaces reach a total separation point. There are various forms of traction–separation law. The similarity amongst them is that each of them describes linear behavior until the damage initiation point. The difference is the damage evolution or softening phase, which may be linear, exponential, polynomial, or bi- or tri-linear, to name a few.

Bi-linear cohesive law (Figure 6a) reasonably models predominantly brittle fracture and matrix cracking in baseline laminates. Significant fiber bridging is observed in veil-interleaved laminates. Such a bridging effect is modeled by modifying the bi-linear to tri-linear law (Figure 6b). The three parameters required to model bi-linear law are cohesive strength (t_m_), penalty stiffness (K), and cohesive energy ($\Gamma_{m}$). An additional three parameters are needed to model tri-linear law, which include bridging strength (t_b_), bridging energy ($\Gamma_{b}$), and the size of the tougher zone (L_a_). These parameters are different and should be determined for each fracture mode. Appendix A shows a collection of parameters used in the literature to model delamination of FRP laminates with bi- and tri-linear cohesive laws.

Bi-linear and tri-linear laws model the delamination of composite laminates [65,66]. Though a significant fiber bridging effect was observed, only bi-linear law was considered to model delamination of the baseline and interleaved laminates. However, it should be noted that bi-linear law only models the first linear part of the traction–separation curve until damage initiation [67,68,69]. The damage evolution part of interleaved laminates cannot be modeled using bi-linear law only; hence the model presented does not reflect the actual behavior.

The FT, experimentally obtained using a DCB [65,66,67] or ENF [65,66,68,69] test, is the initial cohesive energy value for numerical modeling. The cohesive strength and penalty stiffness are calculated using a trial-and-error process. The values of these two parameters are changed and tuned until the numerical model’s force–displacement curve coincides with the experimental curve.

The tuning of parameters starts from penalty stiffness, a numerical parameter chosen to compromise the computational need for the value to be low and the physical requirement for it to be as high as possible. The cohesive strength is tuned until the sharpness of the peak in the experimental force–displacement curve matches with the thus obtained numerical curve. Once these two parameters are adjusted, the FT value is also changed and tuned to match the experimental and numerical softening curve.

A similar trial-and-error principle is followed for tuning bridging strength and bridging energy [65,66] after the bi-linear parameters are obtained until the experimental and numerical softening curves are matched. The size of the tougher zone (L_a_) is only required to model the ENF test. The method to determine L_a_ is not clearly defined.

Numerical modeling was carried using a 2D plane-strain finite element model [65,66,67,68,69]. CPE4R elements were used to model the composite laminate, and COH2D4 elements modeled the cohesive surface in front of the pre-existing crack front. As the size of cohesive elements was sensitive, a size of 0.1–0.2 mm (DCB sample length = 125 mm, ENF sample length = 160 mm) was used to model the cohesive layer. Thus, the obtained cohesive parameters can further be utilized to model the behavior of composite components and parts.

## 8. Conclusions

In this paper, a general trend of improvement in all four parameters used to characterize the FT was observed. Interleaving was proven to be one of the most effective methods for improving the FT of composite laminate. The growing number of publications on this topic indicates that it is in the growth phase of the technology growth curve. The method of production of interleaves, electrospinning, is stable, scalable and well understood, with lots of choices for material selection. Some of the materials that are widely used as interleaves were analyzed and reviewed. The highest attainable FT for various interleave materials was identified and discussed (see Section 6 for comparative analysis). Figure 4 was plotted as a reference for veil material selection as per the FT requirement in the composite design. Various toughening mechanisms introduced by these polymeric interleaves were examined, listed, and discussed. The functionalization of interleaves was discussed, and the key parameters and methodology for the numerical modeling of delamination using cohesive zone modeling was reviewed and outlined.

## Figures and Tables

**Figure 2 polymers-15-01380-f002:**
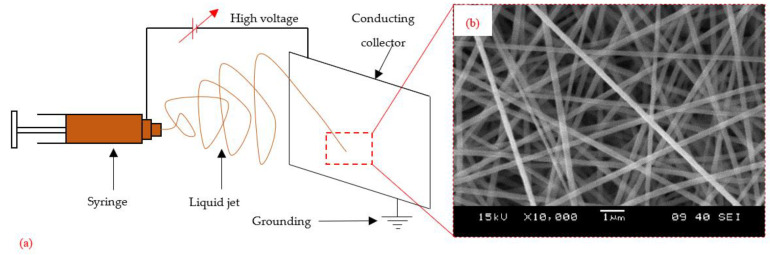
Electrospinning: (**a**) schematic representation; (**b**) a typical PAN veil (courtesy of H. He, K. Molnar, Budapest University of Technology and Economics).

**Figure 3 polymers-15-01380-f003:**
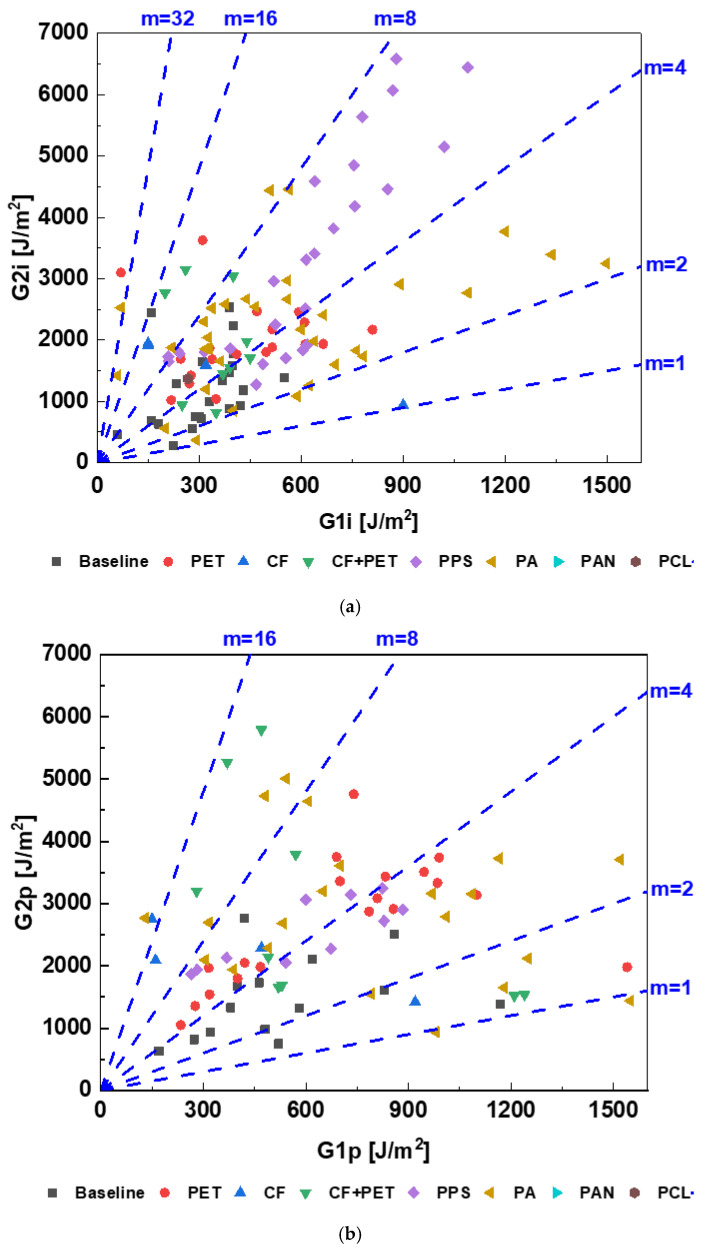
Mode ratio for (**a**) initiation and (**b**) propagation FTs for baseline vs. interleaved laminate. Interleaves of various polymers are considered.

**Figure 5 polymers-15-01380-f005:**
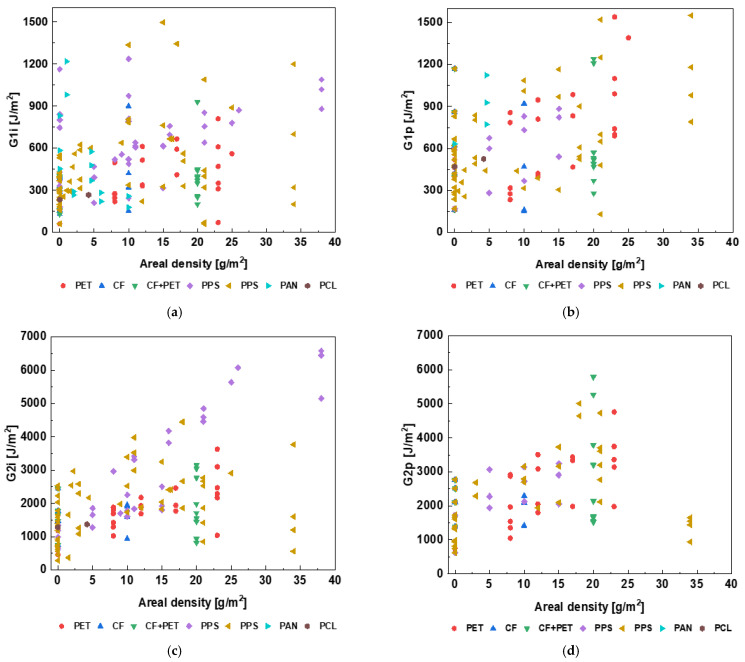
Dependence of (**a**) G1i, (**b**) G1p, (**c**) G2i, and (**d**) G2p on areal density of various electrospun veils. The areal density of 0 means no electrospun veil was placed in the laminate.

**Figure 6 polymers-15-01380-f006:**
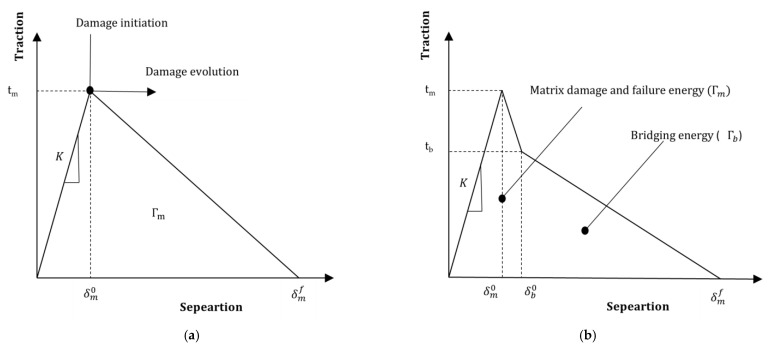
Traction–separation law for cohesive zone modeling: (**a**) bi-linear (**b**) tri-linear cohesive law.

**Table 1 polymers-15-01380-t001:** Correlation between areal density and FT.

	G1i	G1p	G2i	G2p
Correlation coefficient (*r*)	0.44	0.34	0.59	0.58

## Data Availability

The data presented in this study are available on request from the corresponding authors.

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
