# Peer review of "A Review of Electrospun Nanofiber Interleaves for Interlaminar Toughening of Composite Laminates"

_polymers, 2023, doi:10.3390/polym15061380_

Round 1

Reviewer 1 Report

The authors have systematically summarized the electrospun nanofiber interleaves for interlaminar toughening of composite laminates. The corresponding toughening mechanism, statistical analysis and delamination simulation of are also further provided. Before accepting the current paper, the authors are suggested to consider the following comments. 

1. In the first paragraph of the introduction, it is suggested that the authors define the composite material as fiber reinforced polymer composite. According to the type of fiber, it can also be divided into CFRP, BFRP and GFRP, etc. Furthermore, their main performances and advantages should be further emphasized and highlighted, such as excellent mechanical properties, corrosion resistance and creep resistance compared with traditional metallic materials, leading to a wide acceptance and application in engineering. It is suggested to review the following relevant studies and make necessary supplements. CFRP: Composite Structures, 2022, 293, 115719. BFRP: Composite Structures, 2021, 255: 112935. GFRP: Engineering Structures, 2023, 274: 115176.

2. As mentioned, “Various methods are used to improve interlaminar properties of structural compo-sites: matrix toughening [1,2], 3D weaving [3], stitching [3], Z-pinning [4], nano-stitching [5], and hybridization [6].”, it is suggested that the authors analyze the advantages and disadvantages of each improvement method.

3. In part 4, the authors are encouraged to give some figures and tables for the quantitative analysis, so that readers can better understand the toughness effects of different materials.

4. From 4.1-4.5, different materials are used as toughening agent, it is suggested to make relevant comparative summary and analysis on the toughness effects of composites and provide the relevant explanations.

5. It is suggested that the schematic diagram of the toughness mechanism should be given in Part 5.

6. The clarity of figure 3 should be further improved.

7. For the numerical modeling of delamination, is the simulation result verified by the relevant experimental results? How to evaluate the accuracy of the simulation?

8. In the part of the conclusion, please provide the quantitative results about the toughening effect for different toughness materials if possible.

Author Response

polymers-2241530       

Biltu Mahato, Stepan V. Lomov, Aleksei Shiverskii, Mohammad Owais, Sergey G. Abaimov

ANSWERS TO REVIEWERS

Reviewer 1 – For convenience, our revisions are highlighted in yellow.

Reviewer’s comment

Authors’ answer

The authors have systematically summarized the electrospun nanofiber interleaves for interlaminar toughening of composite laminates. The corresponding toughening mechanism, statistical analysis and delamination simulation of are also further provided.

Before accepting the current paper, the authors are suggested to consider the following comments. 

We appreciate a positive evaluation of our analysis and review. We have done the proposed modifications in the revised manuscript.

1. In the first paragraph of the introduction, it is suggested that the authors define the composite material as fiber reinforced polymer composite. According to the type of fiber, it can also be divided into CFRP, BFRP and GFRP, etc. Furthermore, their main performances and advantages should be further emphasized and highlighted, such as excellent mechanical properties, corrosion resistance and creep resistance compared with traditional metallic materials, leading to a wide acceptance and application in engineering. It is suggested to review the following relevant studies and make necessary supplements. CFRP: Composite Structures, 2022, 293, 115719. BFRP: Composite Structures, 2021, 255: 112935. GFRP: Engineering Structures, 2023, 274: 115176.

Composite material as fiber reinforced polymer composite definition. – Added.

Their excellent properties compared to traditional materials leading to a wide acceptance and application in engineering fields. – Added.

Relevant studies done. Necessary supplements considered and added.

2. As mentioned, “Various methods are used to improve interlaminar properties of structural compo-sites: matrix toughening [1,2], 3D weaving [3], stitching [3], Z-pinning [4], nano-stitching [5], and hybridization [6].”, it is suggested that the authors analyze the advantages and disadvantages of each improvement method.

Added and discussed.

3. In part 4, the authors are encouraged to give some figures and tables for the quantitative analysis, so that readers can better understand the toughness effects of different materials.

Comparative explanation added.

4. From 4.1-4.5, different materials are used as toughening agent, it is suggested to make relevant comparative summary and analysis on the toughness effects of composites and provide the relevant explanations.

5. It is suggested that the schematic diagram of the toughness mechanism should be given in Part 5.

Schematic diagram added.

6. The clarity of figure 3 should be further improved.

Figure enlarged.

Line type changed.

Mode ratio moved outside the box for clarity.

7. For the numerical modeling of delamination, is the simulation result verified by the relevant experimental results? How to evaluate the accuracy of the simulation?

Yes, the simulation result must be verified by the relevant experiment. Having experimental results help to achieve the fast convergence in numerical modeling.

The accuracy of simulation is evaluated by checking the match of experimental and simulation force-displacement curve. It has been widely discussed in paragraph 4, 5, 6 of chapter 7.

8. In the part of the conclusion, please provide the quantitative results about the toughening effect for different toughness materials if possible.

Added.

English language editing is highlighted in Pink.

Change in text arrangement to adjust figures is highlighted in Grey.

Reviewer 2 Report

This paper provides an overview of the application of nanofibre polymer veils as a toughened interlayer in fiber-reinforced composite laminates and provides a systematic comparative analysis and summary of the fracture toughness improvements achievable with electrospun veil-based materials. It is of great guidance for polymeric nanofibre veils in industrial applications and is recommended to accept directly.

Author Response

polymers-2241530       

Biltu Mahato, Stepan V. Lomov, Aleksei Shiverskii, Mohammad Owais, Sergey G. Abaimov

ANSWERS TO REVIEWERS

Reviewer 2 – For convenience, our revisions are highlighted in green.

Reviewer’s comment

Authors’ answer

This paper provides an overview of the application of nanofibre polymer veils as a toughened interlayer in fiber-reinforced composite laminates and provides a systematic comparative analysis and summary of the fracture toughness improvements achievable with electrospun veil-based materials. It is of great guidance for polymeric nanofibre veils in industrial applications and is recommended to accept directly.

We appreciate the positive evaluation of our analysis and review.

English language editing is highlighted in Pink.

Change in text arrangement to adjust figures is highlighted in Grey.

Reviewer 3 Report

The main focus of this review paper is on the toughness efficiency of the veils/interleaves produced by electrospinning method. The toughening mechanisms added by introduction of veils are listed, analyzed, and discussed. The polymeric veils are introduced between plies of a composite laminate, and targeted effect on delamination initiation and propagation are broadly studied. This review paper offers an overview of application of nanofiber polymeric veils as toughening interleaves in the fiber-reinforced composite laminates. The numerical modelling of failure by both modes of delamination is also discussed. Present systematic review can be used as guidance for veil material selection, estimation of the achievable toughening effect, understanding toughening mechanism introduced by veils and numerical modelling of delamination.

The review is well-presented, and the topics discussed are interesting, but the authors need to address the following minor issue before acceptance of the paper.

 Minor issue:

1.    The authors should comment in more details improvements on all FT parameters observed in addressed Ref. [28].

Author Response

polymers-2241530            

Biltu Mahato, Stepan V. Lomov, Aleksei Shiverskii, Mohammad Owais, Sergey G. Abaimov

ANSWERS TO REVIEWERS

Reviewer 3 – For convenience, our revisions are highlighted in turquoise.

Reviewer’s comment

Authors’ answer

The main focus of this review paper is on the toughness efficiency of the veils/interleaves produced by electrospinning method. The toughening mechanisms added by introduction of veils are listed, analyzed, and discussed. The polymeric veils are introduced between plies of a composite laminate, and targeted effect on delamination initiation and propagation are broadly studied. This review paper offers an overview of application of nanofiber polymeric veils as toughening interleaves in the fiber-reinforced composite laminates. The numerical modelling of failure by both modes of delamination is also discussed. Present systematic review can be used as guidance for veil material selection, estimation of the achievable toughening effect, understanding toughening mechanism introduced by veils and numerical modelling of delamination.

The review is well-presented, and the topics discussed are interesting, but the authors need to address the following minor issue before acceptance of the paper.

We appreciate a positive evaluation of our analysis and review. We have done the proposed modifications in the revised manuscript.

 Minor issue:

1.    The authors should comment in more details improvements on all FT parameters observed in addressed Ref. [28].

The results for PET veil interleaved laminate in Ref. [28] is discussed in details in last paragraph of chapter 4.1.1.

The results for PA veil interleaved laminate in Ref. [28] is added in chapter 4.3.

English language editing is highlighted in Pink.

Change in text arrangement to adjust figures is highlighted in Grey.

Round 2

Reviewer 1 Report

It is recommended to accept the current paper.